# Potential Correlation between Microbial Diversity and Volatile Flavor Compounds in Different Types of Korean Dry-Fermented Sausages

**DOI:** 10.3390/foods11203182

**Published:** 2022-10-12

**Authors:** Jong-Hui Kim, Eun-Seon Lee, Bu-Min Kim, Mi-Hwa Oh

**Affiliations:** National Institute of Animal Science, Rural Development Administration, Wanju 55365, Korea

**Keywords:** Korean dry-fermented sausage, volatile flavor, microbial diversity

## Abstract

The microbial community in fermented sausages plays an important role in determining their quality characteristics. The objective of this study was to investigate the correlation between microbial diversity and volatile compounds in dry-fermented sausages procured from different regions of Korea. Results from metagenomics analysis showed that *Lactobacillus* and *Staphylococcus* were the predominant bacterial genera, and *Penicillium*, *Debaryomyces*, and *Candida* were the predominant fungal genera. Twelve volatile compounds were detected using an electronic nose. *Leuconostoc* exhibited a positive correlation with esters and volatile flavor, whereas *Debaryomyces*, *Aspergillus*, *Mucor*, and *Rhodotorula* exhibited a negative correlation with methanethiol, thus revealing the involvement of the microorganisms in flavor formation. The results of this study may help in understanding the microbial diversity of dry-fermented sausages in Korea and provide a rationale and quality control guideline through potential correlation with volatile flavor analysis.

## 1. Introduction

Fermentation is an ancient method of preserving meat. During fermentation, the growth of microorganisms causes chemical changes in food that alter its appearance and flavor. The production of fermented sausages involves several steps. Meat is first minced, then mixed with salt, sugar, seasoning, and starter culture; it is then fermented and ripened or dried [1]. Several biochemical and microbiological changes occur during these processes due to the degradation of proteins and lipids [2,3]. Accordingly, many studies have demonstrated that the bacterial community in fermented sausages plays an important role in determining the quality characteristics of sausages.

Starter microbial cultures are used to promote and carry out the fermentation of meat products. They are added to control the growth of spoilage-related microorganisms and standardize the quality characteristics of the product [4]. Lactic acid bacteria (LAB), mainly used as starter cultures, inhibit the growth of pathogenic microorganisms by reducing the pH or producing bacteriocins [5]. In addition, the use of commercial molds on sausage surfaces leads to a consistent flavor, taste, color, and drying rate, as well as a uniform appearance [6]. However, depending on the manufacturing method, instead of starter cultures, microorganisms inherent in the raw material, blend material, and ripening conditions, are used to achieve similar outcomes [7]. In addition, water activity, pH, acidity, nutrient availability, and humidity, as well as the length of the product process, contribute to the establishment of the core microbiome. These parameters have a significant impact on the quality of fermented sausages, and the metabolites therein are believed to provide a characteristic flavor profile [2,5]. However, knowledge of the effects of microbial community diversity and metabolites (volatile and non-volatile) on sausage quality is still lacking [7].

With rapid advances in DNA sequencing techniques and the development of high-throughput sequencing (HTS) technology, specific microbial communities within complex ecosystems can now be characterized. Such findings can contribute to our understanding of flavor and quality in foods such as fermented sausages [8,9]. Furthermore, recent studies have investigated the potential correlation between core bacteria and volatile flavor compounds in fermented sausages [7,8]. However, although several papers have been published on the microbiome of fermented sausages, most of them are limited to Europe. To date, most of the studies have focused on bacterial communities based on the 16S rRNA sequence. Therefore, our understanding of the diversity and role of eukaryotes in fermented sausages is limited [4,8,9]. Both bacterial and eukaryotic ecosystems present in Korean fermented sausages should be analyzed using HTS technology to obtain a more accurate assessment of their microbial diversity and quality.

In this study, HTS was used to understand the composition of bacterial and fungal communities and the key microorganisms contributing to volatile flavor compounds, and to evaluate the safety of sausages by identifying spoilage-related microorganisms. In addition, the physiochemical parameters of dried sausages, including water content, water activity (a_w_), and pH, were analyzed. The results of this study could help in producing safe, high-quality, dry-fermented sausages, and improving their quality and consistency.

## 2. Materials and Methods

### 2.1. Preparation of Sausage Samples

Dry-fermented sausage samples (labeled A, B, C, and D) were procured from meat production factories located in four different regions of Korea. The sausages had been produced according to the following process. We compiled the ingredients (listed in Table 1) of each sample based on the manufacturer’s instructions. Following homogenization in a vacuum mixer, the mixture was inoculated with LAB (except sample C). Subsequently, the mixture was filled into a natural pig or artificial casing to form a cylindrical shape. The casing surface was then inoculated with a mold starter culture (except sample D). Sausages were hung with a string and fermented at 23–27 °C with a relative humidity (RH) of 100%. Fermentation was carried out until a final pH was about 5 (within approximately 48 h). Subsequently, the fermented sausages were dried until the weight decreased by approximately 35% at 13–15 °C, RH of 75–80%. Samples were individually packaged in sterile bags, placed in a container with dry ice, and transferred to the laboratory within 12 h. Table 1 lists the specific features of each dried fermented sausage sample.

### 2.2. Estimation of Physicochemical Characteristics of Sausage Samples

#### 2.2.1. Proximate Composition, pH, and Water Activity (a_w_)

The protein, fat, moisture, and collagen contents of the fermented sausages were analyzed using the Food Scan^TM^ Lab 78,810 (Foss Tecator Co. Ltd., Hillerod, Denmark). Briefly, approximately 200 g of each sample was homogenized with a stomacher blender (Bag Mixer 400, Interscience, St. Nom la Breteche, France) for 2 min. Then, the sample was placed in the sample dish of the Food Scan and loaded into the sample chamber. The pH of fermented sausages was measured using a pH meter. Water activity (a_w_) of the fermented sausages was determined at 25 °C with a Novasina measuring instrument, model AW SPRINT-TH 300 (Pfaffikon, Switzerland). All experiments were performed in triplicate.

#### 2.2.2. Color Measurement

The color characteristics of the sausage samples were measured using the Minolta Chroma Meter CR-400 (Minolta Co. Ltd., Osaka, Japan). The color parameters assessed included lightness (L*), redness (a*), and yellowness (b*). The equipment was standardized with a white color standard.

### 2.3. HTS of Sausage DNA

#### 2.3.1. DNA Extraction

Approximately 25 g of each sausage sample was homogenized in 225 mL of sterile 0.1% peptone water (Oxoid, Cambridge, UK) for 10 min in a stomacher. The samples were then centrifuged at 10,000× *g* for 20 min at 4 °C, and the supernatants (1 mL) were transferred to new tubes for DNA extraction using the DNeasy Blood and Tissue kit (Qiagen, Hilden, Germany), following the manufacturer’s protocols.

#### 2.3.2. Illumina HTS

Next generation sequencing (NGS) was used to characterize bacterial 16S rDNA gene and the fungal ITS region. Primers were designed to amplify the V3–V4 region of the 16S rDNA gene and the ITS2 region (ITS3-4 primer). Amplicon libraries were prepared using the 16S Metagenomic Sequencing Library preparation kit. The Herculase II Fusion DNA polymerase Nextera XT Index Kit V2 was used to prepare 16S rRNA gene amplicons in the Illumina MiSeq System. Polymerase chain reaction (PCR) was performed in a final volume of 25 μL, which contained 12.5 μL of 2 × KAPA HiFi HotStart ReadyMix (Roche, Basel, Switzerland), 5 μL of 1 μM primers, and 12 ng of genomic DNA. A two-step PCR was performed for the amplification. The first PCR step consisted of an initial denaturation at 95 °C for 3 min; this was followed by 25 cycles of denaturation at 95 °C for 30 s, annealing at 55 °C for 30 s, and elongation at 72 °C for 30 s; and a final extension at 72 °C for 5 min. The final PCR step consisted of an initial denaturation at 95 °C for 3 min, followed by eight cycles of denaturation at 95 °C for 30 s, annealing at 55 °C for 30 s, elongation at 72 °C for 30 s, and a final extension at 72 °C for 5 min.

The quality and quantity of the PCR product were measured using the Quant-IT™ PicoGreen^®^ kit and a Bioanalyzer 2100, respectively. Paired-end sequencing for 600 cycles was performed using the Illumina MiSeq sequencing platform (GnCBIO Company, Daejeon, Korea), according to the manufacturer’s standard protocol.

#### 2.3.3. Data Analysis

Raw gene sequence data were analyzed using the open-source QIIME 2 pipeline [10,11]. First, the sequences were quality filtered using the DADA2 algorithm in QIIME 2. A masked alignment using MAFFT was then conducted, followed by the construction of a phylogeny tree of the sequences using the QIIME 2 FastTree plugin. The 16S sequences were assigned to taxonomic groups using the SILVA database (https://www.arb-silva.de/) (accessed on 10 January 2021) as a reference.

Sequences with more than 97% similarity were clustered into operational taxonomic units (OTUs) for the next step of bioinformatic analysis. To evaluate bacterial richness and diversity, alpha diversity indices, including ACE, Chao1, Shannon, Simpson, Good’s coverage, and rarefaction analysis, were analyzed using QIIME. Non-metric multidimensional scaling (NMDS) and cluster analysis were performed in R (https://www.r-project.org/) (accessed on 10 January 2021) using Vegan, ggplot 2 packages, and the Jaccard distance method at the genus level. Edges were assigned to samples with a similarity of >0.70.

### 2.4. Aroma and Taste Profiling

#### 2.4.1. Volatile Compound Profiling Using an Electronic Nose

Aroma profiles of the four sausage samples were analyzed using a Heracles II electronic nose (Alpha MOS, Toulouse, France) according to the method of Go et al. [12], with minor modifications. Homogenized sausage samples (5 g) were placed in a 20 mL vial and heated at 60 °C for 20 min. Subsequently, 1 mL of each sample was delivered by the autosampler to the injector at 125 μL/s at 200 °C. The sample was maintained for 20 s at 40 °C in a TENAX absorbent trap. The two columns (MXT-5/MXT-1701, Restek, Bellefonte, PA, USA) were mounted, and the detector temperature was maintained at 260 °C. Subsequently, the distinguished principal component (aroma substance) of the samples was considered as the primary component (PC1) and secondary component (PC2) values. The AlphaSoft software (Alpha MOS, Toulouse, France) was used for classified aroma profiling.

#### 2.4.2. Taste Profiling Using an Electronic Tongue

Taste profiles of the four sausage samples were analyzed using a potentiometric electronic tongue (Astree V, Alpha MOS) according to the method used by Go et al. [12], with minor modifications. Then, 5 g of each sample was weighed and homogenized for 1 min in 25 mL of distilled water using a homogenizer. The homogenate was filtered using filter paper, and the filtrate was diluted 100 times in a glass container. The water phase was equipped with five chemical sensors for umami (NMS), saltiness (CTS), sourness (AHS), sweetness (PKS), and bitterness (ANS), as well as general-purpose sensors (SCS and CPS).

### 2.5. Statistical Analysis

Physicochemical data are expressed as mean ± standard deviation, and differences between samples were analyzed using one-way analysis of variance. Differences between means were assessed using Duncan’s multiple range tests using IBM SPSS Statistics version 26. Differences were considered significant at *p* < 0.05. Three independent batches of dry fermented-sausages (replicates) were collected, and all measurements were conducted in triplicate (triplicate observations) for each batch of dry fermented-sausage. A biplot based on principal component analysis (PCA) was constructed using the XLSTAT software package version 2021 (Addinsoft, Paris, France). Spearman’s correlation analysis was performed using IBM SPSS (v26) to determine the relationship between the microbial community and the volatile compounds identified using the e-nose and e-tongue. Correlations were determined using Spearman’s correlation coefficient.

## 3. Results

### 3.1. Physicochemical Properties of the Sausage Samples

Differences in the manufacturing methods and ingredients of dry fermented sausages may lead to differences in their physicochemical properties. The physicochemical properties, including moisture, fat, protein, collagen, a_w_, pH, and color, of the dry-fermented sausage samples from four different regions of Korea are listed in Table 2. The moisture content and a_w_ of the sausage samples ranged from 27.39–30.96% and 0.78–0.85, respectively. Sample D, which had the largest diameter, had the highest moisture content and a_w_ (*p* < 0.05). The pH values differed significantly among the four samples (*p* < 0.05); sample B had the lowest pH at 3.62. The color values were significantly different among the four samples (*p* < 0.05); L* (lightness), a* (redness), and b* (yellowness) ranged from 38.22–46.58, 8.62–12.80, and 6.49–9.03, respectively.

### 3.2. Microbial Richness and Diversity

Illumina MiSeq sequencing was performed to characterize the bacterial and fungal diversity and community structure of the dry-fermented sausage samples. Based on the sequencing results, 1,071,389 high-quality 16S rRNA gene sequences and 752,104 high-quality ITS2 region sequences were identified from the four dry-fermented sausage samples. These sequence reads were clustered into 149 OTUs at a 97% similarity level per sample, of which 87 OTUs belonged to bacteria, and 62 OTUs were attributed to fungi. The alpha diversity indices, including Chao1, Shannon, Simpson, and God’s coverage analyses, were evaluated for the four samples (Table 3). Results showed that sample B had the highest value for Chao 1, followed by samples A, D, and C. Bacterial diversity (Shannon and Simpson indices) was also highest in sample B, followed by A, D, and C. By contrast, sample A had the highest fungal richness and diversity indices. Good’s coverage values were 100% for both bacterial and fungal OTUs in all the samples.

### 3.3. Similarity of Microbial Communities

Based on the NMDS analysis at the genus level, the differences in microbial distribution between samples were reflected in the distance between points. Clustering analysis also supported the results of the NMDS sorting diagram, and the distance between the microbial communities at each region is shown. As shown in Figure 1a, the sausage samples were separated into two different clusters based on the bacterial community. Samples A, C, and D were grouped in one cluster, whereas sample B formed another cluster. Similarly, cluster analysis using the Vegdist function of the Vegan package indicated that bacterial communities of samples A, C, and D were relatively more similar to each other than to those of sausage sample B (Figure 1c). As shown in Figure 1b, the NMDS results at the fungal genus level revealed two different clusters; sausage samples A, B, and C were grouped in one cluster, whereas sample D formed another cluster. Cluster analysis also supported this result (Figure 1d).

### 3.4. Composition of the Microbial Communities

The relative abundances of the bacterial and fungal communities at the phylum and genus levels are shown in Figure 2. For the bacterial communities, 3 phyla, 4 classes, 6 orders, 12 families, and 14 genera were identified. *Firmicutes* and *Proteobacteria* were the predominant phyla in all the samples, accounting for over 99% of all OTUs (Figure 2a). *Actinobacteriota* (0.9%) was present only in sample B. At the genus level, *Lactobacillus* (96.4%) was the most abundant in the all the sausage samples, except for sausage B (Figure 2b). The community composition of sample B was slightly different from that of the other sausage samples. Specifically, *Pediococcus* (53.2%) was more dominant than *Lactobacillus* (31.1%), and a total of 11 genera, including *Staphylococcus, Weissella, Leconostoc, Carnobacterium, Brochothrix, Pseudomonas, Psychrobacter, Serratia*, and *Sphingomonas*, were observed in sample B. On the other hand, the bacterial community of sausage C without LAB starter culture was composed of *Lactobacillus* (97.3%) and *Staphylococcus* (2.7%).

Fungal OTUs were clustered into 3 phyla, 7 classes, 10 orders, 14 families, and 18 genera. At the phylum level (Figure 2c), *Ascomycota* was the most abundant in all the samples, accounting for more than 97.9% of all OTUs, whereas *Basidiomycota* represented a small portion (0.3–2.1%). *Mucoromycota* was detected in only sample D (1.5%). At the genus level (Figure 2d), five to seven fungal taxa were detected in all the samples. *Penicillium* was the most abundant in samples A (51.6%), B (75.1%), and C (86.5%), whereas *Debaryomyces* (91.8%) was the most abundant in sample D. Other major genera included *Candida*, *Wickerhamomyces*, *Aspergillus*, *Dipodascus*, and *Fusarium*, with relative abundances of >1%.

### 3.5. E-Nose and E-Tongue Analysis

Twenty-two volatile compounds were detected in the four sausage samples using an e-nose, including two ketones (protane, 2-propanone), two esters (ethyl acetate, trichlorethylene), two sulfur-containing compounds (methanethiol; 2,4,5-trimethylthiazole), hydrocarbons (cyclopentane, 4-methylnonane), one furan (2-methylfuran), and other compounds (limonene) (Figure 3a). High levels of ethanol were detected in all the samples.

As shown in Figure 3b, sample A had the highest AHS value, which was consistent with the pH results. The highest ANS, SCS, and CPS values were observed in sample B. Sample D exhibited high levels of NMS and CTS, whereas sample C exhibited medium levels of all values.

Data collected from the aroma and taste profiling experiments were elaborated using PCA (Figure 3c). PC1 accounted for the highest proportion of variance (58.15%), which was associated with the differences between the four sausage samples. The first group (encircled in red) corresponds to sausage sample D and is characterized by 4-methylnonane, 2,4,5-trimethylthiazole, cyclopentane, and limonene flavors. The second group (encircled in blue) consists of samples A and C and is distinguished by propanone, ethanol, 2-methylfuran, ethyl acetate, thiophene, and trichloroethylene compounds. The third group (encircled in green) consists of sample B, is characterized by pentane, and is associated with all taste sensors except saltiness.

### 3.6. Potential Correlation of Microbiota, Flavors, and Tastes

Bacterial and fungal communities may play important roles in the volatile flavor development of fermented sausages. Therefore, a correlation analysis was conducted to reveal the relationship between microbial composition and volatile compounds of the sausage samples using Pearson’s correlation analysis. As shown in the PCA loading plots (Figure 4), the clustering trend of the four sausage samples was still observable, indicating differences among the samples with regard to microbiota, aroma, and taste. PC1 (45.36%) distinguished sample B, which exhibited a positive correlation from samples A, C, and D. Samples A, B, and C were placed on the negative axis of PC2, whereas sample D exhibited a positive correlation with PC2 (33.81%).

As shown in Table 4, eight bacterial genera and six fungal genera exhibited a significant correlation with flavor (*p* < 0.05). Methanethiol, a sulfur-containing compound, exhibited a negative correlation with *Debaryomyces*, *Aspergillus*, *Mucor*, and *Rhodotorula* (*p* < 0.05); no significant correlation was observed with bacteria. Pentane exhibited a positive correlation with *Brichothrix*, *Pseudomonas*, *Psychrobacter*, *Serratia*, *Dipodascus*, and *Pichia*, (*p* < 0.05) and was negatively correlated with *Lactobacillus* (*p* < 0.01). In addition, 2-propanone, ethyl acetate, and 2-methylfuran were positively correlated with *Leuconostoc*, and 2,4,5-trimethylthiazole with *Staphylococcus*, respectively (*p* < 0.05). Sourness, bitterness, SCS, and CPS were positively correlated with *Brichothrix*, *Pediococcus*, *Pseudomonas*, *Psychrobacter*, *Serratia*, *Dipodascus*, and *Pichia*, whereas umami was negatively correlated with *Lactobacillus* and positively correlated with *Trichosporon* (*p* < 0.05). Sweetness and saltiness were not significantly associated with any microorganism.

## 4. Discussion

Dry-fermented sausage manufacturing is a well-known biotechnology technique performed by natural fermentation or initiated by a LAB starter culture and/or coagulase-negative staphylococci (CNS) [13]. However, species diversity of microbial communities within the fermented meat matrices is known to depend on the raw materials used, ripening and drying environment, and processing method applied [2,3,5]. Therefore, detailed information on the composition and diversity of the microbial community is necessary. The present study investigated the physicochemical properties of dry-fermented sausages from four different regions of Korea and established potential correlations between the core microbiome, major volatile compounds, and taste.

Depending on the optimal growth conditions, physicochemical and biochemical changes that occur during the ripening process in the manufacture of dry-fermented sausages lead to the development of specific microbial populations [3]. The a_w_ levels of the samples were in the order of D > A = C > B (*p* < 0.05). The higher a_w_ of sample D could be due to its higher water content and larger diameter. The lowest moisture content (27.39%) and a_w_ (0.78) were observed in sample B, which had the smallest section diameter (26 mm). Significant differences in pH values were observed among the four sausage samples. These results indicate that the LAB content distribution in the sausages resulted in stronger acidification. Consistently, in bacterial NGS analysis, sample C exhibited the highest level of *Lactobacillus* (99.9%), whereas sample B had the lowest *Lactobacillus* abundance (31.3%). High *Lactobacillus* levels in sample C, which was not inoculated with LAB starter culture, could have resulted from the spontaneous generation of LAB with high acidification power [14]. In sausage manufacturing, the required final pH of the finished product is 5.3 or lower. This is an important technical quality characteristic that reflects the shelf-life stability of fermented meat products to be considered “storage stable” [15]. However, the pH value of sample B (pH 5.87) seemed insufficient to guarantee the shelf-life stability of the product, and accordingly, genera associated with spoilage, such as *Serratia*, *Pseudomonas*, and *Brochothrix,* were detected.

The structure of the microbial communities, including bacterial and fungal diversity, of the four dry-fermented sausage samples, was investigated using HTS technology. The bacterial core microbiome was predominantly LAB (*Lactobacillus*, *Lactococcus*, *Weissella*, *Pediococcus*, and *Leuconostoc*), accounting for 86–99.9% of the total 16S rRNA. Unexpectedly, 99.9% of the *Lactobacillus* was observed in sample C, which had not been inoculated with a starter culture. By contrast, in Italian-style dry sausages prepared using a traditional method without the addition of starter cultures, *Staphylococcaceae* was the dominant bacterium and reached a relative abundance of up to 97.52%, whereas that of *Lactobacillales* was only 1.72% [16]. However, LAB is generally detected at higher levels than CNS and is better adapted to changes during the fermentation process [17,18]. This distribution difference may be due to the difference in the salinity of the sausages. *Staphlococcus* spp. is naturally well-adapted to high salinity conditions, and *Staphlococcus equorum* and *Staphylococcus xylosus* have been found in brine for the maturation of meat [19,20]. In fermented sausages, lactobacilli metabolize carbohydrates, and thus exert a great influence on flavors, which are mainly produced by the resulting alcohols and ketones [21]. Many lactobacilli can produce esterases [22] that cause the formation of sweet and fruity esters by the esterification of carboxylic acids with alcohol [23]. Accordingly, in the present study, ethanol and 2-propanone derived from carbohydrate metabolism, and esters (ethyl acetate and trichloroethylene) produced by bacterial esterification were positively correlated with *Leuconostoc* (*p* < 0.05).

*Staphylococcus* was detected at a fairly low abundance of 1.8–3.9% in all the samples and positively correlated with 2,4,5-trimethylthiazole (*p* < 0.05). A previous study reported that 2,4,5-trimethylthiazole is a sulfur-containing (S-containing) heterocyclic compound that is mainly found in meat products, such as chicken broth, cooked beef, and roasted lamb fat [24,25]. S-containing compounds are derived from the degradation of S-containing amino acids, such as methionine, cysteine, and cystine [14]. CNS has been reported to exhibit high protease and lipase activity and plays a considerable role in the flavor development of fermented meat by producing low molecular weight compounds [26].

*Serratia*, *Pseudomonas*, and *Brochothrix* were detected at 5.5%, 1.1%, and 0.5%, respectively, in sample B alone, and the genus will require species identification in future research. *Serratia*, along with *Pseudomonas*, is the most common genus present on work surfaces in the meat processing industry [18] and is the main spoilage factor for fresh, cooked, or cured meat. The active growth, acidification, and proteolytic capacity of these microbial populations provide favorable conditions for biogenic amine production during meat ripening [27]. Biogenic amines, when consumed, have several adverse effects on health, including the development of allergies, hypertension, and headaches; hence, they are considered the determinants of poor-quality meat products [27]. Significant concentrations of biogenic amines, such as putrescine and cadaverine, are often observed in meat products, and *Enterobacteriaceae*, such as *Serratia*, are thought to be responsible for the formation of these amines [28]. In particular, *Serratia* produces large quantities of cadaverine and putrescine with growth in vitro [27]. *Pseudomonas aeruginosa* is a pathogen that predominantly spreads in patients with burns, cystic fibrosis, and acute leukemia, and in post-organ transplantation or intravenous drug poisoning [29]. *Brochothrix thermosphacta* is a facultative anaerobic bacterium that can contaminate the entire process of fermented meat production, from raw material to final product, and impart an off-flavor to the meat [30]. Therefore, controlling the presence of *Serratia*, *Pseudomonas*, and *Brochothrix* in dry-fermented meat can contribute to the production of superior-quality products [31]. In this study, the results of correlation analysis between volatile compounds and bacteria were very similar between Spearman’s ranking method and PCA. *Serratia*, *Pseudomonas*, and *Brochothrix* to sample B were positively correlated with pentane, a ketone (*p* < 0.05). Pentane has been reported to be a typical undesirable off-flavor compound [32].

*Penicillium* spp. is the most commonly used fungus starter in meat products such as dry-cured ham or dry-fermented sausages [6]. As expected, in this study, *Penicillium* was the dominant species in samples A, B, and C, with *Penicillium nalgiovense* as the dominant species (76–96.8%) (Appendix A). In addition, *P. bialowiezense* (0.1%) and *P. neocrassum* (0.1%) were detected in sample C. *P. verucosum* and *P. nordicum*, which are capable of producing ochratoxins in dried fermented sausages, were not detected [33].

*Debaryomyces*, identified as *Debaryomyces prosopidis* at the species level (Appendix A), was the most abundant genus (92.4%) in sample D. Similar results were found in smoked bacon stored for 120 d, where *D. prosopidis* accounted for 90.05% [34]. *D*. *prosopidis* is predominantly present in dry-fermented meat products and contributes positively to the development of sensory traits [34,35]. *Candida*, along with *Debaryomyces*, is the most widely isolated yeast in meat products [36]. In this study, *Candida* was detected in all the sausage samples (6.8–35.3%), except in sample D. However, depending on the species, caution should be exercised in increasing its population. *Candida zeylanoides* inhibits the growth of *Penicillium nordicum* in dried ham and dried fermented sausages, and also plays an important role in improving the flavor and quality of meat products [37]. However, *Candida parapsilosis*, detected in sample B (1.1%), is a yeast that causes sepsis, as well as wound and tissue infections in immunocompromised individuals [38]. This species is also a normal human commensal and is one of the most frequently isolated fungi from human hands [38]. *Aspergillus ruber*, which is abundant in Meju and spices, was detected in sample D (3.9%). However, caution should be exercised in increasing its population because it can produce aflatoxin B1 and ochratoxin A [39].

The fungal communities of the sausage samples were largely divided into two groups: *Penicillium* (samples A, B, and C) and *Debaryomyces* (sausage sample D). Surprisingly, *Penicillium* spp., which is known to contribute to product flavor due to its high proteolytic and lipolytic activity [6,40] was not significantly correlated with volatile flavors. By contrast, *Debaryomyces* showed a negative correlation with rancid methanethiol (*p* < 0.05) and no significant positive correlation with other microorganisms. Methanethiol has the characteristic odor of rotten cabbage similar to sewage and produces a pungent odor in spoiled ham. [41]. Conversion into these compounds may lead to consumer acceptance or rejection of the product. Therefore, an abundance of *Debaryomyces* could help reduce off-flavors. A negative correlation with methanethiol was also observed in *Aspergillus*, *Mucor*, and *Rhodotorrula* (*p* < 0.05), which play important roles in the fermentation of sausages through their protease and lipase production capacity. These fungi break down proteins into small molecule polypeptides, and amino acids and lipids into glycerol and fatty acids, which positively affects the color and flavor of products [34,42].

## 5. Conclusions

Our metagenomic analysis revealed the presence of significant microbial diversity in dry-fermented sausages procured from four different regions of Korea. *Lactobacillus* and *Staphylococcus* were the predominant bacterial genera, whereas *Penicillium*, *Debaryomyces*, and *Candida* were the predominant fungal genera. *Leuconostoc* showed a positive correlation with ester, and *Debaryomyces, Aspergillus, Mucor*, and *Rhodotorula* showed a negative correlation with methanethiol, which plays an important role in flavor development. Additionally, the pH of sausage products can cause outgrowths of potentially harmful bacteria, particularly members of *Enterobacterales*. Future research should elucidate the function and fermentation mechanism of core microorganisms beneficial to the development of the volatile flavor of dried sausage. Furthermore, changes in volatile and nonvolatile compounds due to microbial changes during sausage fermentation should be explored. These factors can help speed up the fermentation process and standardize the quality of industrial dried sausages.

## Figures and Tables

**Figure 1 foods-11-03182-f001:**
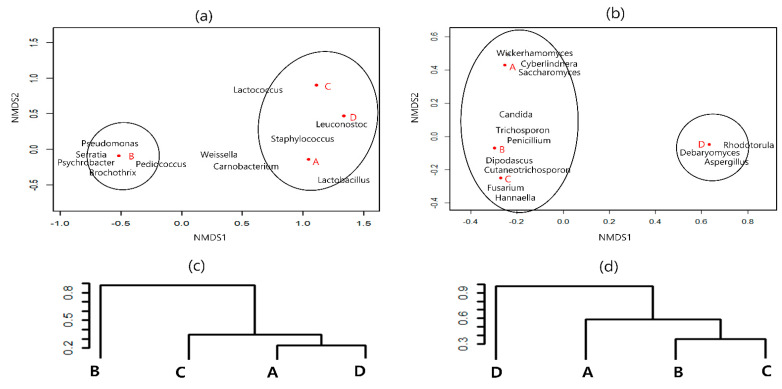
Nonmetric multidimensional scaling (NMDS) analysis of bacterial (**a**) and fungal (**b**) communities of the dry−fermented sausage samples. Circles are the area in which each sausage is most closely related to microbial population. Cluster analysis of bacterial (**c**) and fungal (**d**) communities of the dry−fermented sausage samples. Dist.mat (Euclidean distance matrix between points)/Confidence cutoff = 0.70.

**Figure 2 foods-11-03182-f002:**
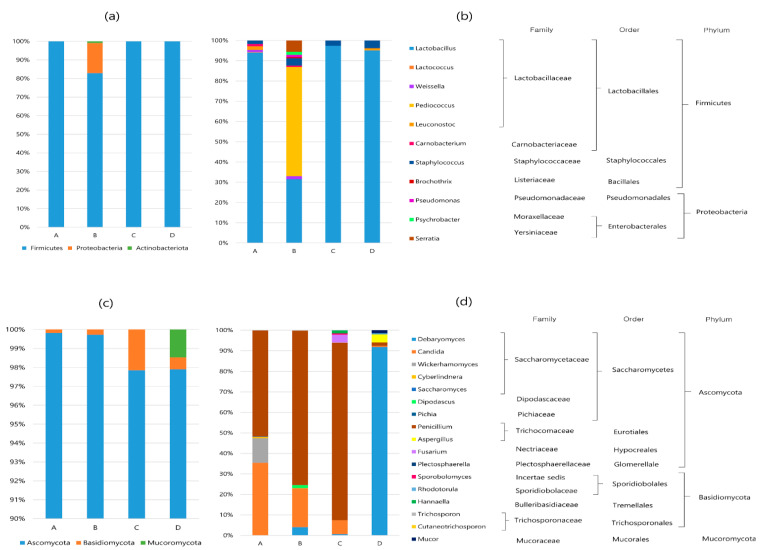
Relative abundance (%) of the bacterial communities at the phylum (**a**) and genus (**b**) levels, and relative abundance (%) of the fungal communities at the phylum (**c**) and genus (**d**) levels in dry-fermented sausage samples from four different regions of Korea (cut off: ≥0.1% abundance).

**Figure 3 foods-11-03182-f003:**
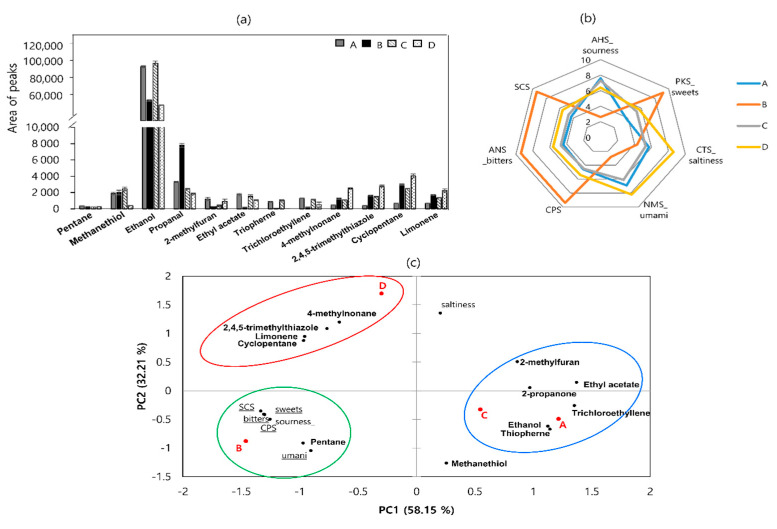
Levels of volatile compounds detected using an electronic nose (**a**), radar chart using an electronic tongue (**b**), and principal component analysis (PCA) loading plot (**c**) of dry−fermented sausage samples from four different regions of Korea. Circles are the areas in which each sausage is most closely related to volatile flavors.

**Figure 4 foods-11-03182-f004:**
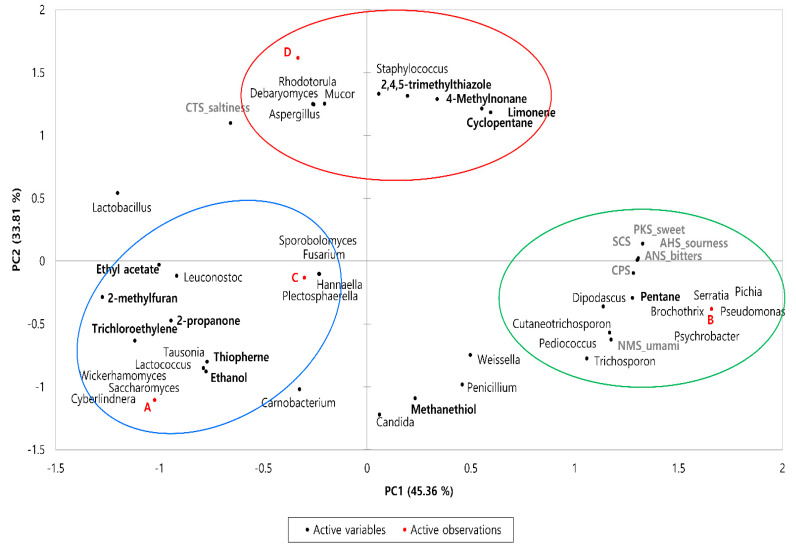
PCA loading plots between the microbiome, volatile compounds, and taste of dry−fermented sausage samples from four different regions of Korea. Circles are the areas in which the microbial population is most closely related to volatile flavors.

**Table 1 foods-11-03182-t001:** Manufacturing information of dry-fermented sausage samples from different regions of Korea.

Sample	Starter Culture		Ingredient List	Fermentation Period
	Lactic Acid Bacteria (LAB)	Mold		
A	*Lactiplantibacillus plantarum*	*Penicillium nalgiovens*	Pork shoulders, pork fat, white sugar, refined salt, onion, garlic, ginger, pepper powder, kelp, shiitake mushroom, yeast extract, vegetable mixed powder, barbecue flavor seasoning, collagen casing	40–45 d
B	*Pediococcus acidilactici, Latilactobacillus curvatus*, *Staphylococcus xylosus*	*Penicillium nalgiovens*	Pork shoulders, pork fat, pepper powder, glucose, garlic, sodium nitrite, natural (pig intestine) casing	25–30 d
C	Non-starter	*Penicillium nalgiovens*	Pork shoulders, pork fat, red wine, salt, black pepper, garlic, nitrite, ascorbic acid	40–45 d
D	Mixed LABs (21 species)	Non-starter	Pork shoulders, pork fat, slat 1%, herb mix (coriander, clove, fennel, oregano), garlic, black pepper, salt, nitrate	60–65 d

**Table 2 foods-11-03182-t002:** Physicochemical analysis of dry-fermented sausage samples from four different regions of Korea.

	Sausage Products			
Parameter	A (*n* = 3)	B (*n* = 3)	C (*n* = 3)	D (*n* = 3)
pH	4.70 ± 0.19 ^c^	5.87 ± 0.03 ^a^	3.62 ± 0.08 ^d^	5.05 ± 0.07 ^b^
a_w_	0.82 ± 0.00 ^b^	0.78 ± 0.00 ^a^	0.82 ± 0.01 ^b^	0.85 ± 0.01 ^a^
Moisture (%)	29.31 ± 0.31 ^b^	27.39 ± 0.15 ^c^	28.93 ± 0.16 ^b^	30.96 ± 0.30 ^a^
Fat (%)	33.22 ± 0.59 ^b^	29.46 ± 0.44 ^c^	34.38 ± 0.42 ^a^	32.71 ± 0.60 ^b^
Protein (%)	35.28 ± 0.15 ^b^	38.12 ± 0.13 ^a^	34.38 ± 0.33 ^c^	34.23 ± 0.11 ^c^
Collagen (%)	2.59 ± 0.03 ^b^	4.43 ± 0.15 ^a^	2.15 ± 0.04 ^c^	2.51 ± 0.06 ^b^
L* value	46.58 ± 1.36 ^a^	38.22 ± 1.33 ^c^	40.37 ± 1.51 ^b^	47.05 ± 1.58 ^a^
a* value	8.62 ± 1.22 ^c^	11.62 ± 0.69 ^b^	12.80 ± 0.61 ^a^	12.77 ± 0.69 ^a^
b* value	9.03 ± 0.63 ^a^	6.49 ± 0.98 ^c^	8.04 ± 0.66 ^b^	8.88 ± 0.26 ^a^
Section diameter (mm)	40	26	40	60

^a,b,c^ Means with different uppercase letters were significantly different (*p* < 0.05).

**Table 3 foods-11-03182-t003:** The operational taxonomic units (OTUs) of dry-fermented sausages from four different regions of Korea.

Sample	Reads		Observed OTUs		Chao 1 Index		Good’s Coverage		Shannon Index		Simpson	
	Bacteria	Fungi	Bacteria	Fungi	Bacteria	Fungi	Bacteria	Fungi	Bacteria	Fungi	Bacteria	Fungi
A	420,848	156,310	28	14	28	14	1.0	1.0	4.10	2.03	0.92	0.68
B	131,765	193,906	36	12	45	12	1.0	1.0	4.43	1.27	0.95	0.40
C	135,926	218,900	6	24	15	12	1.0	1.0	2.24	1.37	0.87	0.45
D	382,850	182,988	17	12	19	12	1.0	1.0	3.53	1.32	0.90	0.44

Richness estimators (Chao 1) and diversity indices (Shannon and Simpson) of the bacterial 16S rRNA and fungal ITS2-region for each sample were determined using OTU-based analysis.

**Table 4 foods-11-03182-t004:** Spearman’s correlation coefficients between microbial composition, flavor, and taste of dry-fermented sausage samples from four different regions of Korea.

Variables	Volatile Flavor												Taste Profile						
	2-Propanone	Methanethiol	Ethanol	Pentane	2-Methylfuran	Ethyl acetate	Thiopherne	Trichloroethylene	4-Methylnonane	2,4,5-Trimethylthiazole	Cyclopentane	Limonene	AHS_ Sourness	PKS_ Sweets	CTS_ Saltiness	NMS_ Umami	ANS_ Bitters	CPS	SCS
**Bacteria**																			
*Lactobacillus*	0.398	−0.402	0.331	−0.997 **	0.564	0.805	0.363	0.604	0.256	0.173	−0.022	−0.002	−0.909	−0.855	0.735	−0.971 *	−0.91	−0.936	−0.906
*Carnobacterium*	0.809	0.254	0.357	0.249	0.595	0.276	0.268	0.407	−0.737	−0.83	−0.857	−0.798	−0.115	−0.314	−0.34	0.097	−0.109	−0.021	−0.13
*Brochothrix*	−0.486	0.243	−0.502	0.974 *	−0.591	−0.906	−0.525	−0.748	−0.061	0.02	0.21	0.196	0.973 *	0.932	−0.601	0.932	0.974 *	0.986 *	0.971 *
*Pediococcus*	−0.486	0.243	−0.502	0.8	−0.591	−0.906	−0.525	−0.748	−0.061	0.02	0.21	0.196	0.973 *	0.932	−0.601	0.932	0.974 *	0.986 *	0.971 *
*Leuconostoc*	0.958 *	−0.443	0.112	−0.399	0.994 **	0.959 *	0.051	0.425	−0.124	−0.295	−0.469	−0.366	−0.516	−0.684	0.469	−0.619	−0.51	−0.463	−0.527
*Pseudomonas*	−0.486	0.243	−0.502	0.974 *	−0.591	−0.906	−0.525	−0.748	−0.061	0.02	0.21	0.196	0.973 *	0.932	−0.601	0.932	0.974 *	0.986 *	0.971 *
*Staphylococcus*	−0.351	−0.838	−0.707	−0.376	−0.018	−0.265	−0.653	−0.524	0.924	0.972 *	0.898	0.925	0.061	0.146	0.816	−0.434	0.059	−0.016	0.07
*Lactococcus*	0.946 *	0.148	0.526	−0.132	0.784	0.601	0.45	0.666	−0.672	−0.791	−0.887	−0.827	−0.475	−0.648	−0.095	−0.259	−0.469	−0.391	−0.488
*Weissella*	0.375	0.64	0.004	0.748	0.144	−0.289	−0.08	−0.095	−0.625	−0.653	−0.568	−0.529	0.455	0.272	−0.611	0.602	0.461	0.538	0.443
*Psychrobacter*	−0.486	0.243	−0.502	0.974 *	−0.591	−0.906	−0.525	−0.748	−0.061	0.02	0.21	0.196	0.973 *	0.932	−0.601	0.932	0.974 *	0.986 *	0.971 *
*Serratia*	−0.486	0.243	−0.502	0.974 *	−0.591	−0.906	−0.525	−0.748	−0.061	0.02	0.21	0.196	0.973 *	0.932	−0.601	0.932	0.974 *	0.986 *	0.971 *
**Fungal**																			
*Penicillium*	−0.518	0.528	0.506	0.394	−0.756	−0.086	0.53	0.151	−0.56	−0.408	−0.243	−0.352	0.199	0.303	−0.852	0.646	0.194	0.201	0.203
*Debaryomyces*	−0.037	−0.963 *	−0.675	−0.465	0.297	−0.117	−0.645	−0.399	0.93	0.853	0.729	0.79	−0.076	−0.051	0.935	−0.598	−0.076	−0.133	−0.071
*Candida*	0.538	0.936	0.383	0.521	0.252	0.072	0.301	0.29	−0.861	−0.893	−0.839	−0.816	0.123	−0.046	−0.663	0.437	0.128	0.215	0.11
*Wickerhamomyces*	0.946	0.148	0.526	−0.132	0.784	0.601	0.45	0.666	−0.672	−0.791	−0.887	−0.827	−0.475	−0.648	−0.095	−0.259	−0.469	−0.391	−0.488
*Aspergillus*	−0.018	−0.959 *	−0.641	−0.505	0.316	−0.072	−0.609	−0.358	0.92	0.842	0.711	0.771	−0.121	−0.094	0.949	−0.633	−0.121	−0.179	−0.117
*Fusarium*	−0.451	0.572	0.621	−0.343	−0.517	0.38	0.689	0.443	−0.185	−0.068	−0.029	−0.138	−0.381	−0.191	−0.254	−0.041	−0.387	−0.421	−0.37
*Dipodascus*	−0.486	0.243	−0.502	0.974 *	−0.591	−0.906	−0.525	−0.748	−0.061	0.02	0.21	0.196	0.973 *	0.932	−0.601	0.932	0.974 *	0.986 *	0.971 *
*Mucor*	−0.011	−0.962 *	−0.647	−0.498	0.322	−0.077	−0.616	−0.363	0.918	0.839	0.708	0.77	−0.115	−0.091	0.948	−0.629	−0.115	−0.172	−0.111
*Hannaella*	−0.449	0.571	0.622	−0.345	−0.515	0.382	0.69	0.445	−0.185	−0.068	−0.03	−0.139	−0.383	−0.194	−0.253	−0.043	−0.389	−0.423	−0.373
*Sporobolomyces*	−0.449	0.571	0.622	−0.345	−0.515	0.382	0.69	0.445	−0.185	−0.068	−0.03	−0.139	−0.383	−0.194	−0.253	−0.043	−0.389	−0.423	−0.373
*Rhodotorula*	−0.011	−0.962 *	−0.647	−0.498	0.322	−0.077	−0.616	−0.363	0.918	0.839	0.708	0.77	−0.115	−0.091	0.948	−0.629	−0.115	−0.172	−0.111
*Cyberlindnera*	0.946	0.148	0.526	−0.132	0.784	0.601	0.45	0.666	−0.672	−0.791	−0.887	−0.827	−0.475	−0.648	−0.095	−0.259	−0.469	−0.391	−0.488
*Trichosporon*	−0.469	0.714	0.01	0.898	−0.706	−0.595	−0.006	−0.339	−0.472	−0.351	−0.14	−0.2	0.729	0.726	−0.921	0.984 *	0.728	0.754	0.728
*Saccharomyces*	0.946	0.148	0.526	−0.132	0.784	0.601	0.45	0.666	−0.672	−0.791	−0.887	−0.827	−0.475	−0.648	−0.095	−0.259	−0.469	−0.391	−0.488
*Tausonia*	0.946	0.148	0.526	−0.132	0.784	0.601	0.45	0.666	−0.672	−0.791	−0.887	−0.827	−0.475	−0.648	−0.095	−0.259	−0.469	−0.391	−0.488
*Cutaneotrichosporon*	−0.781	0.615	−0.102	0.758	−0.929	−0.665	−0.082	−0.466	−0.181	−0.024	0.192	0.107	0.733	0.814	−0.769	0.911	0.729	0.719	0.738
*Pichia*	−0.486	0.243	−0.502	0.974 *	−0.591	−0.906	−0.525	−0.748	−0.061	0.02	0.21	0.196	0.973 *	0.932	−0.601	0.932	0.974 *	0.986 *	0.971 *
*Plectosphaerella*	−0.449	0.571	0.622	−0.345	−0.515	0.382	0.69	0.445	−0.185	−0.068	−0.03	−0.139	−0.383	−0.194	−0.253	−0.043	−0.389	−0.423	−0.373

* The correlation was significant at *p* < 0.05; ** The correlation was significant at *p* < 0.01.

## Data Availability

The data presented in this study are available in the article.

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
