# Peer review of "Potential Correlation between Microbial Diversity and Volatile Flavor Compounds in Different Types of Korean Dry-Fermented Sausages"

_foods, 2022, doi:10.3390/foods11203182_

Round 1
Reviewer 1 Report
General comments:
The authors introduced too many sausage preparation variables into the experiment. These variables significantly distort the real picture of the chemicals being analyzed, especially during fermentation. This makes it impossible to draw generalized conclusions.
The ingredients used in sausages A, B, C, and D vary greatly from one another. The variations relate to the source of the meat (e.g., the raw material's unknown chemical properties), seasonings (which vary amongst sausages), and additions (e.g., sodium nitrate, ascorbic acid). They undoubtedly influence the environment in which microflora grows. Chemicals can have both positive and negative, synergistic or antagonistic impacts, which are not disregarded in this. The authors should prove that different ingredients and process parameters do not affect the formation of flavor substances. Otherwise, the results have very limited scientific significance.
On the other hand, they well characterized four selected Korean dry-fermented sausages in terms of the presence of volatile flavor compounds, indicating the presence of specific microorganisms after processing (fermentation). This can be accepted if the title of the work states that different sausages were used.
Detailed comments:
line 25 - In addition to the ingredients listed, starter cultures are also used.
Line 68 - Not all sausages in the experiment have such casings (list of ingredients in Table 1).
Table 1 - replace the uppercase letter with a lowercase letter in the word "Mix";
Parenthetical 2.3.2 should be completed with supplier / manufacturer data (e.g. MiSeq, PicoGreen, Bioanalyzer, etc.).
Line 117 - Redundant hyphen in the word "Illumina".
line 120 - Necessary reference to the source (QIIME 2). The correct notation is QIIME 2.
line 129 - Replace the lowercase letter with an uppercase letter in the word "non-...".
line 136 - Necessary information about the manufacturer of e-nose.
line 139 - Necessary information on the manufacturer MXT.
line 145 - Necessary information on the manufacturer of e-tongue.
Figure 2 - Some colors are very difficult to distinguish on the chart. Please improve the readability of the information.
Author Response
General comments:
The authors introduced too many sausage preparation variables into the experiment. These variables significantly distort the real picture of the chemicals being analyzed, especially during fermentation. This makes it impossible to draw generalized conclusions.
The ingredients used in sausages A, B, C, and D vary greatly from one another. The variations relate to the source of the meat (e.g., the raw material's unknown chemical properties), seasonings (which vary amongst sausages), and additions (e.g., sodium nitrate, ascorbic acid). They undoubtedly influence the environment in which microflora grows. Chemicals can have both positive and negative, synergistic or antagonistic impacts, which are not disregarded in this. The authors should prove that different ingredients and process parameters do not affect the formation of flavor substances. Otherwise, the results have very limited scientific significance.
On the other hand, they well characterized four selected Korean dry-fermented sausages in terms of the presence of volatile flavor compounds, indicating the presence of specific microorganisms after processing (fermentation). This can be accepted if the title of the work states that different sausages were used.
Response: Thank you for your thoughtful feedback. We agree with your comments. Currently, there are four types of commercially produced sausages in Korea, all of which were used in this study. We analyzed the physicochemical, sensory and microbiological characteristics of these fermented sausages. We propose that the title be changed to:
Evaluation of microbial diversity and volatile flavor compounds in different types of Korean dry-fermented sausages
In addition, the demand for Korean-style fermented sausages in the country is increasing, and some small meat processing companies are attempting to fulfill these needs. In line with these demands, it is necessary to investigate the microbial diversity and major flavor components of various types of Korean fermented sausages. To the best of our knowledge, the microbial diversity of Korean sausages has not yet been investigated using high-throughput sequencing.
Detailed comments:
- line 25 - In addition to the ingredients listed, starter cultures are also used.
Response: We have now mentioned starter cultures in our ingredient list.
- Line 68 - Not all sausages in the experiment have such casings (list of ingredients in Table 1) Response: We have revised the sentence to accommodate this exception.
- Table 1 - replace the uppercase letter with a lowercase letter in the word "Mix"
Response: We have revised the word. - Parenthetical 2.3.2 should be completed with supplier / manufacturer data (e.g. MiSeq, PicoGreen, Bioanalyzer, etc.). Response: We have included the details of the supplier.
- Line 117 - Redundant hyphen in the word "Illumina".
Response: We have deleted the hyphen. - line 120 - Necessary reference to the source (QIIME 2). The correct notation is QIIME 2. Response: We have included the reference.
- line 129 - Replace the lowercase letter with an uppercase letter in the word "non-...".
Response: We have revised the letter to the uppercase. - line 136 - Necessary information about the manufacturer of e-nose.
Response: We have included this information in the revised version of the manuscript. - line 139 - Necessary information on the manufacturer MXT.
Response: We have included this information in the revised version of the manuscript. - line 145 – Necessary information on the manufacturer of e-tongue.
Response: We have included this information in the revised version of the manuscript. - Figure 2 - Some colors are very difficult to distinguish on the chart. Please improve the readability of the information.
Response: We have changed ambiguous colors in Figure 2.
Reviewer 2 Report
This study was to investigate the correlation of microbial diversity and volatile flavor compounds from 4 factorie-made Korean dry-fermented sausages. Overall, the research was properly conducted and the data was well-interpreted. One major concern is the sampling size when performing the correlation analysis with only using 3 sausages from each factory. Thus the result need to be verified. Another concern is the sausage origin, which need to be regarded as a factor when performing correlation analysis. The authors take so much effort in comparing the physicochemical parameters, microbial communities, and flavors in sausage among 4 factories, while the manufacturing processes are different, in despite of starter culture. The correlation analysis between microbial communities and flavors does not suggest that this is a cause effect.
Other suggestions:
Line 45-48 Please add recent findings of microbiome in fermented sausage, which is associated meat quality, especially flavor compounds. Thus, the authors could rephrase the innovation of this study, other than only outlining the origin of sausage.
Line 63 how many sausages were obtained for each labeling, i.e A, B, C, D treatment. How many times were the samples for each purchasing, one time sampling from each factory ? Thus, please specify the replicate of the samples, factory or sausage ?
Line 152-154 Please add the detection replicate of microbial community and volatile compounds using the e-nose and e-tongue. Simple Pearson’s correlation analysis is not adequate in the current study because there is a factor (sausage origin) involved in the analysis.
Author Response
This study was to investigate the correlation of microbial diversity and volatile flavor compounds from 4 factory-made Korean dry-fermented sausages. Overall, the research was properly conducted and the data was well-interpreted. One major concern is the sampling size when performing the correlation analysis with only using 3 sausages from each factory. Thus the result need to be verified. Another concern is the sausage origin, which need to be regarded as a factor when performing correlation analysis. The authors take so much effort in comparing the physicochemical parameters, microbial communities, and flavors in sausage among 4 factories, while the manufacturing processes are different, in despite of starter culture. The correlation analysis between microbial communities and flavors does not suggest that this is a cause effect.
Response: Thank you for your thoughtful feedback. We agree with your comments. Currently, there are four types of commercially produced sausages in Korea, all of which were used in this study. We analyzed the physicochemical, sensory and microbiological characteristics of these fermented sausages. We propose that the title be changed to:
Evaluation of microbial diversity and volatile flavor compounds in different types of Korean dry-fermented sausages
In addition, the demand for Korean-style fermented sausages in the country is increasing, and some small meat processing companies are attempting to fulfill these needs. In line with these demands, it is necessary to investigate the microbial diversity and major flavor components of various types of Korean fermented sausages. To the best of our knowledge, the microbial diversity of Korean sausages has not yet been investigated using high-throughput sequencing.
Other suggestions:
- Line 45-48 Please add recent findings of microbiome in fermented sausage, which is associated meat quality, especially flavor compounds. Thus, the authors could rephrase the innovation of this study, other than only outlining the origin of sausage. Response: Thank you for the comment. We have included a sentence about the recent findings of microbiome in fermented sausage.
- Line 63 how many sausages were obtained for each labeling, i.e A, B, C, D treatment. How many times were the samples for each purchasing, one time sampling from each factory? Thus, please specify the replicate of the samples, factory or sausage? Response: We have specified the replicate of the samples and included them in the subsection on statistical analysis in the Materials and Methods section.
- Line 152-154 Please add the detection replicate of microbial community and volatile compounds using the e-nose and e-tongue.
Response: We have included replicates of samples used for analysis. - Simple Pearson’s correlation analysis is not adequate in the current study because there is a factor (sausage origin) involved in the analysis.
Response: Thank you for your suggestion. We have determined the relationship between the microbial community and volatile compounds using Spearman's correlation analysis. Minor changes have been made in the text to accommodate this change.
Reviewer 3 Report
General information.
The topic of the review article fits within the scope of the journal, and the article is well written. The Authors have provided deep insight into microbial communities and the aroma profile of native Korean meat products. The correlation between specific microorganisms, aroma, and compound formation is always welcome since it is very interesting for the development of foods with increased nutritional value.
Please address the following:
Introduction
Page 1, line 24-25. Please rewrite the sentence so that the order of fermented sausage is clear. The meat is first minced then mixed with salt sugar and seasoning and then fermented.
Page 1, lines 45-48. Please rewrite the sentence to be more consistent.
2.1. Preparation of sausage samples
Page 2, lines 63-64. The analyzed sausages were produced based on manufacturers' recipes, or were those recipes compiled by authors? Please clarify that.
2.4.1. Volatile compound profiling using an electronic nose
Page 4, the whole paragraph, do you have any reference for the method applied?
2.4.2. Taste profiling using an electronic tongue.
Same as for 2.4.1, any reference?
2.5. Statistical analysis
What was the difference level of significance, p-value?
3.4. Composition of the microbial communities
Is there any more information on microbial communities for sample C?
3.6. Correlation analysis of microbiota, flavors, and tastes
Page 10. Table 4. It seems that some parts of the table are missing please correct the formatting.
4. Discussion
Page 12, line 341-343. Could that be due to some missteps in production?
Page 12, Line 346-347. Are those biogenic amines potentially toxic, or linked to in vitro cytotoxicity? Please, elaborate on why where those biogenic amines mentioned in the text, for the broader audience to understand.
Which microorganisms had a positive correlation with methanethiol formation? For the broader audience please elaborate on why is methanethiol so specific, and why the limitation of its production is important.
References
Please add the DOI where is it possible and check the reference list again, there are a few minor corrections needed.
I presume that the authors will continue with the research and use more specific recipes, and compile some new and beneficial ones for the production of distinct aroma profiles. The following course of research could be noted.
Round 2
Reviewer 2 Report
The authors have properly addressed my comments.